# CT Texture Analysis of Adrenal Pheochromocytomas: A Pilot Study

**Filippo Crimì** [1,2], **Elena Agostini** [1,2], **Alessandro Toniolo** [1,2], **Francesca Torresan** [3,4], **Maurizio Iacobone** [3,4], **Irene Tizianel** [1,5], **Carla Scaroni** [1,5], **Emilio Quaia** [1,2], **Cristina Campi** [6] and **Filippo Ceccato** [1,5,*]

1   Department of Medicine (DIMED), University of Padova, Via Giustiniani 2, 35128 Padova, Italy
2   Institute of Radiology, Padova University Hospital, Via Giustiniani 2, 35128 Padova, Italy
3   Department of Surgical, Oncological and Gastroenterological Sciences, University of Padova, Via Giustiniani 2, 35128 Padova, Italy
4   Endocrine Surgery Unit, Padova University Hospital, Via Giustiniani 2, 35128 Padova, Italy
5   Endocrinology Unit, Padova University Hospital, Via Giustiniani 2, 35128 Padova, Italy
6   Department of Mathematics (DIMA), University of Genova, Via Dodecaneso 33, 16146 Genova, Italy
*   Correspondence: filippo.ceccato@unipd.it

**Abstract:** Radiomics is a promising research field that combines big data analysis (from tissue texture analysis) with clinical questions. We studied the application of CT texture analysis in adrenal pheochromocytomas (PCCs) to define the correlation between the extracted features and the secretory pattern, the histopathological data, and the natural history of the disease. A total of 17 patients affected by surgically removed PCCs were retrospectively enrolled. Before surgery, all patients underwent contrast-enhanced CT and complete endocrine evaluation (catecholamine secretion and genetic evaluation). The pheochromocytoma adrenal gland scaled score (PASS) was determined upon histopathological examination. After a resampling of all CT images, the PCCs were delineated using LifeX software in all three phases (unenhanced, arterial, and venous), and 58 texture parameters were extracted for each volume of interest. Using the Mann–Whitney test, the correlations between the hormonal hypersecretion, the malignancy score of the lesion (PASS > 4), and texture parameters were studied. The parameters DISCRETIZED_HUpeak and GLZLM_GLNU in the unenhanced phase and GLZLM_SZE, CONVENTIONAL_HUmean, CONVENTIONAL_HUQ3, DISCRETIZED_HUmean, DISCRETIZED_AUC_CSH, GLRLM_HGRE, and GLZLM_SZHGE in the venous phase were able to differentiate secreting PCCs ($p < 0.01$), and the parameters GLZLM_GLNU in the unenhanced phase and GLRLM_GLNU and GLRLM_RLNU in the venous differentiated tumors with low and high PASS. CT texture analysis of adrenal PCCs can be a useful tool for the early identification of secreting or malignant tumors.

**Keywords:** adrenal pheochromocytoma; computed tomography; radiomics; texture analysis; LifeX; malignancy; hormonal hypersecretion

## 1. Introduction

Pheochromocytomas (PCCs) and paragangliomas are rare neuroendocrine tumors that originate from chromaffin cells producing catecholamines; in around 40% of cases, PCCs show association with hereditable syndromes [1–3]. The endocrine diagnosis of PCCs is made with the measurement of plasma-free or urinary fractionated metanephrines. The symptoms of these catecholamine-producing tumors include sweating, headache, anxiety, palpitations, and tremors; however, to one-fourth of the patients are asymptomatic at diagnosis [1]. In 30–60% of cases, PCCs are discovered as adrenal incidentalomas in a diagnostic examination performed for an aim other than the study of hypertension or adrenal diseases [4]. Adrenal incidentaloma is a common condition in the general population, especially in those with hypertension or cardiovascular events [5]. Therefore, it is of utmost importance to look for a rare condition (such as PCCs: their prevalence in

patients with hypertension in outpatient clinics ranges between 0.2 and 0.6% [1]) among patients with a high-prevalence disease (such as adrenal incidentaloma: its prevalence is up to 10% in the elderly) [6].

In the case of biochemical evidence of disease, guidelines suggest performing computed tomography (CT) to identify the PCC [1]. Unfortunately, radiological appearance of PCCs is very heterogenous; PCC has been called "the imaging chameleon" [7]. Moreover, the local aggressiveness of PCCs can be classified based on histopathology results, usually according to the pheochromocytoma of the adrenal gland scaled score (PASS) [8]. According to PASS classification, PCCs can be divided in less aggressive or benign forms, the major damages of which in catecholamine-producing tumors are those related to high blood pressure or, in malignant forms, that are more prone to develop distant metastases [9]. In clinical practice, the distinction between benign and malignant PCCs based only on histological criteria is not dogma; there are no certain histological features to define a lesion as malignant (the overt sign of malignancy is the presence of local infiltration or metastatic lesions).

Different imaging modalities have been tested for the identification of benign or malignant subtypes of PCCs, but none of them has been able to exactly predict the probability of malignancy of PCCs before histopathological examination [10]. Some studies have shown a difference in metaiodobenzylguanidine (MIBG) uptake between malignant and benign PCCs, with malignant PCCs exhibiting higher uptake [11,12].

It is of paramount importance to discern between benign and malignant PCCs with CT imaging because, as such discrimination can inform clinical management or surgical approaches. A large lesion that is correctly classified as non-suspect upon preoperative imaging can be treated with conservative surgery (e.g., laparoscopic approach) instead of performing a more demolitive operation.

The main application studied for texture analysis in adrenal lesions in the literature is the identification of benign or malignant adrenal tumors. In a recent review, the pooled area under the receiver operating characteristic curve for the differentiation between benign from malignant lesions was 0.85, showing the high diagnostic accuracy of the technique for this purpose [13]. Moreover, different studies have shown that CT texture analysis can be used to differentiate PCCs from lipid-poor adenomas [14,15]. Recently, a pilot study tested the differences between CT textural parameters in benign and malignant PCCs and paragangliomas [16]. Moreover, Ansquer et al. [17] studied the application of 18F-FDG PET/CT radiomics features in the exclusive characterization of PCCs and their genetic background. Other research groups studied the application of CT texture analysis in the characterization of adrenal masses including the histological subtypes PCCs; one group used texture analysis to distinguish between benign/malignant and functioning/non-functioning adrenal lesions [18], a second group focused on the differentiation between adrenal metastases and benign adrenal lesions [19], and two other groups found that CT texture analysis is useful to differentiate PCCs from lipid-poor adenomas [14,15].

Therefore, starting from these data and from the fact that CT scans are widely available, our aim with this pilot study was to study the application of CT texture analysis in PCCs. The secondary aims are to analyze the correlation between the extracted features and the secretory, histopathological, and behavioral features.

## 2. Materials and Methods

### 2.1. Patients

This retrospective study was approved by the local ethics committee. Between February 2013 and February 2020, we retrospectively retrieved all patients that were operated on for adrenalectomy and received a definitive diagnosis upon histopathological examination of PCCs at the University of Padova. Recruited patients had to present the following criteria: (1) be evaluated at our third-level referral hospital and have undergone adrenal contrast-enhanced CT, including unenhanced and contrast-enhanced scans, with at least 5 mm slice thickness for unenhanced scan and 3 mm slice thickness for arterial and ve-

nous phases scans; (2) have a complete panel of hormonal secretion; (3) have received a histopathologically confirmed diagnosis of PCC after adrenalectomy and, if possible, the evaluation of the PASS [8]. The only exclusion criteria were (1) poor-quality CT images and (2) motion or breathing artifacts.

### 2.2. Image Analysis

A 64-slice CT scanner (Somatom Sensation, Siemens Healthineers, Erlangen, Germany) was used for imaging. The protocol included unenhanced, arterial (15 s after the achievement of 100 HU within the abdominal aorta lumen), and venous-phase (80 s after intravenous contrast injection) acquisitions after intravenous injection of 2 mL/kg of iohexol 350 mg I/mL (Omnipaque, GE Healthcare, Milwaukee, WI, USA) followed by a 50 mL saline flush. The slice thickness reconstruction was 5 mm for unenhanced scans and 3 mm for arterial and venous-phase scans.

Each CT scan was retrieved from the institutional archive system, anonymized, and loaded on a dedicated workstation, where it was analyzed with an independently developed open-access image analysis software for texture analysis (LIFEx, Local Image Features Extraction, Orsay, France) [20]. All CTs were resampled to a voxel size of $1 \times 1 \times 3$ mm (X spacing, Y spacing, Z spacing). Two abdominal radiologists (15 and 5 years of experience) blinded to clinical and histopathological data, identified the PCCs, and an ROI was manually drawn along the tumor in each axial slice of each unenhanced, arterial, and venous phase (Figure 1). A volume of interest (VOI) for each tumor was then obtained in each phase. LIFEx software was used to analyze the voxels within the entire VOIs and compute a set of textural parameters for each of them.

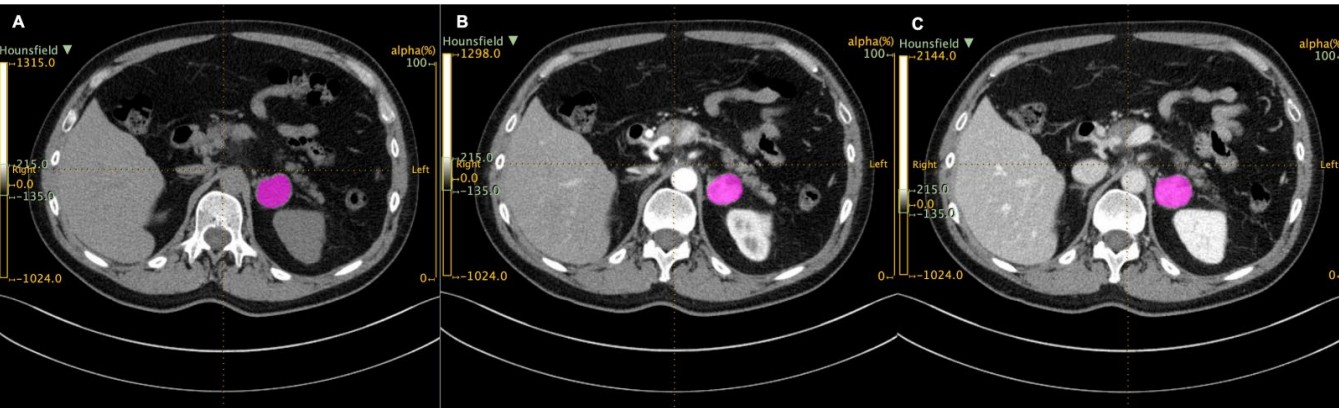

**Figure 1.** Regions of interest (ROIs, in purple) drawn with LIFEx in unenhanced (**A**), arterial (**B**), and venous (**C**) scans.

### 2.3. Statistical Analysis

A total of 58 radiomic features were extracted from the densitometry data (their complete description is reported in Supplementary Table, available in repository data) of both first and second order. First-order statistics describe the distribution of pixels in the VOI using histograms, whereas second-order statistics describe how many neighboring pixels have the same gray level and their relationship in the image. Data were calculated for each PCC in each phase, and groups for secretion and PASS suggestive of benign (PASS $\leq$ 4) or malignant (PASS > 4) behavior were compared with the Mann–Whitney test if not normally distributed and Student's *t* test if normally distributed with Bonferroni correction. The level of significance was set to $p < 0.01$. Statistical analysis was performed using R statistical software (version 2.14.0; R Foundation for Statistical Computing, Vienna, Austria).

### 3. Results

A total of 17 patients (4 females and 13 males; mean age, 48 years; maximum age, 73; minimum age, 26; IQR, 20) affected by histologically confirmed PCC were retrospectively enrolled (a complete description of patients is reported in Table 1). Catecholamine secretion was extensively studied in the urinary collection in all cases, and the PASS scoring system was available in 16 lesions. Of 17 patients, 11 were symptomatic at the time of diagnosis. The most frequent clinical manifestations included secondary hypertension, headache, and sweating. Additionally, 15 of 17 patients had elevated hormonal levels in the measurement of urinary metanephrine, normetanephrine, or catecholamines, reflecting hypersecretion. There was no match between the impaired hormone status and the clinical manifestations. After genetic analysis, two tumors were found to be syndromic. An SDHB mutation was reported in the first, with a TMEM 127 mutation in the second.

**Table 1.** Clinical data of the analyzed patients.

| Case, Gender | Age | Adrenal Size | Genetic Testing | PASS Score | Catecholamine Secretion |
|---|---|---|---|---|---|
| 1, male | 53 | 79 mm | wild type | 5 | MN, NMN |
| 2, male | 41 | 86 mm | wild type | 9 | NMN |
| 3, male | 51 | 33 mm | SDHB mutation | 0 | NA |
| 4, female | 49 | 30 mm | wild type | 1 | MN, NMN |
| 5, male | 28 | 72 mm | wild type | 12 | A, NA, MN, NMN |
| 6, female | 28 | 40 mm | TMEM mutation | 1 | A, NA, MN, NMN, VMA |
| 7, male | 51 | 32 mm | wild type | 2 | NS |
| 8, male | 47 | 54 mm | wild type | 4 | NA, DA, NMN |
| 9, female | 63 | 36 mm | wild type | 0 | A, NA, NMN |
| 10, male | 28 | 30 mm | wild type | 1 | A, DA |
| 11, male | 63 | 38 mm | wild type | 13 | DA |
| 12, male | 51 | 150 mm | wild type | n.a. | A, NA, DA, MN, NMN |
| 13, male | 43 | 80 mm | wild type | 2 | NMN |
| 14, male | 72 | 28 mm | wild type | 0 | A, NA, NMN |
| 15, female | 62 | 19 mm | wild type | 1 | NS |
| 16, male | 30 | 54 mm | SDHB VUS | 1 | NA, NMN, VMA |
| 17, male | 75 | 25 mm | wild type | 1 | A, NA, NMN, VMA |

VUS: variants of uncertain significance; PASS: pheochromocytoma of the adrenal gland scaled score; NS: not secreting; A: adrenaline; NA: noradrenaline; MN: metanephrine; NMN: normetanephrine; VMA: vanilmandelic acid; DA: dopamine.

A total of 13 patients underwent laparoscopic surgery (one retroperitoneal laparoscopic intervention), whereas 3 patients needed an open laparotomic approach. The histopathological data of the resected lesions showed an average PASS score of 2.9, with a minimum value of 0 and a maximum value of 12 (IQR 2.5). Of 17 resected tumors, 5 showed an elevated risk of malignancy (PASS > 4 or qualitative anatomopathological report of high risk in the case in which the PASS score had not been calculated).

In the comparison between the catecholamine-secreting and non-secreting PCC groups (reported in Figure 2 and in Supplementary Material in the repository data: https://doi. org/10.25430/researchdata.cab.unipd.it.00000741, accessed on 25 December 2022) the parameter that showed a statistically significant difference in the unenhanced phase was DISCRETIZED_HUpeak (discretized Hounsfield unit peak), whereas in the venous phase, the parameters that showed a statistically significant difference were GLZLM_GLNU (gray-level zone length matrix–gray-level non-uniformity), GLZLM_SZE (gray-level zone length matrix–short-zone emphasis), CONVENTIONAL_HUmean (conventional Hounsfield unit mean), CONVENTIONAL_HUQ3 (conventional Hounsfield unit third quartile), DISCRETIZED_HUmean (discretized Hounsfield unit mean), DISCRETIZED_AUC_CSH (discretized area under the curve of the cumulative histogram), GLRLM_HGRE (gray-level run-length matrix–high gray-level run emphasis), and GLZLM_SZHGE (gray-level zone length matrix–short-zone high gray-level emphasis).

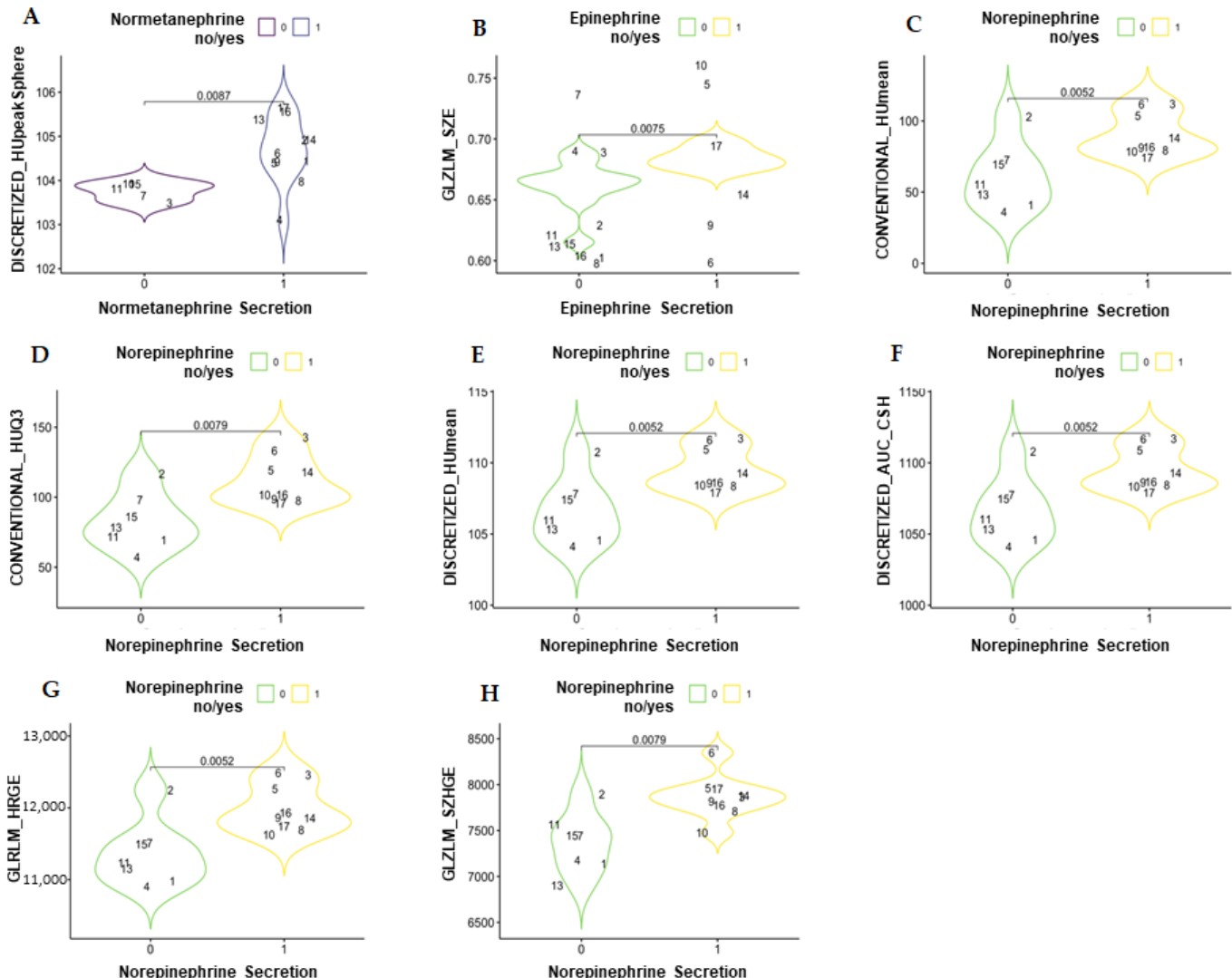

**Figure 2.** Violin plots showing the statistically significant differences in textural parameters between secreting and non-secreting pheochromocytomas in an unenhanced scan (**A**) and in venous scans (**B**–**H**). DISCRETIZED_HUpeak = discretized Hounsfield unit peak; GLZLM_ GLNU = gray-level zone-length matrix–gray-level non-uniformity; GLZLM_SZE = gray-level zone length matrix–short-zone emphasis; CONVENTIONAL_HUmean = conventional Hounsfield unit mean; CONVENTIONAL_HUQ3 = conventional Hounsfield unit third quartile; DIS-CRETIZED_HUmean = discretized Hounsfield unit mean; DISCRETIZED_AUC_CSH = discretized area under the curve of the cumulative histogram; GLRLM_HGRE = gray-level run-length matrix– high gray-level run emphasis; GLZLM_SZHGE = gray-level zone length matrix–short-zone high gray-level emphasis.

In the comparison between high and low PASS grading, GLZLM_GLNU (gray-level zone-length matrix–gray-level non-uniformity) in the unenhanced phase and GLRLM_GLNU (gray-level run-length matrix–gray-level non-uniformity) and GLRLM_RLNU (gray-level run-length matrix–run-length non-uniformity) in the venous phase showed a statistically significant difference between the groups (Figure 3 and Supplementary Table in the repository data: https://doi.org/10.25430/researchdata.cab.unipd.it.00000741, accessed on 25 December 2022). The results were calculated in all three phases (unenhanced, arterial, and venous) and are completely reported in the Supplementary Material available in repositories.

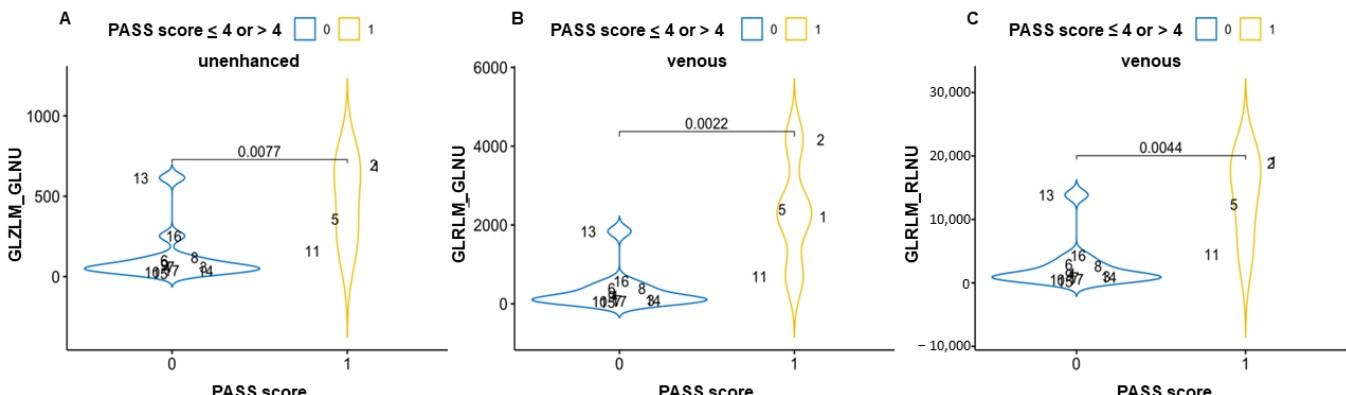

**Figure 3.** Violin plots showing the statistically significant differences in textural parameters between pheochromocytomas with PASS score ≤ 4 or >4 in an unenhanced scan (**A**) and venous scans (**B**,**C**). GLZLM_GLNU = gray-level zone-length matrix–gray-level non-uniformity; GLRLM_GLNU = gray-level run-length matrix–gray level non-uniformity; GLRLM_RLNU = gray-level run-length matrix–run-length non-uniformity.

## 4. Discussion

The present study suggests that textural features extracted from unenhanced and contrast-enhanced CT scans are a promising non-invasive tool to differentiate between hypersecretive and non-hypersecretive and benign from malignant PCCs. Texture analysis on CT images has proven useful, especially in differentiating benign from malignant adrenal masses [13].

Recently, the application of 18F-FDG PET/CT radiomics features was used to characterize PCCs and their genetic background [17] or to differentiate PCCs from lipid-poor adenomas [14,15]. Another study evaluated the correlation between laboratory results, histopathological findings, and features extracted from texture analysis of CT scans in a cohort of PCCs. Regarding the histopathological characteristics, the study showed a correlation between PASS score and the volume of the lesion (pshapevolvx parameter) [21]. A second study found that CT textural features significantly differed between benign and malignant PCCs and paragangliomas (tumor intensity textures such as spatial correlation, laws, LoG, and Gabor and tumor local surface shape, such as shape index SI7) [16]. Likewise, we observed that texture analysis can play a role not only in the identification of catecholamine-producing and non-producing PCCs but also in distinguishing PCCs with a higher or lower PASS.

Observing the parameters that were significantly different between secreting and not secreting PCCs, we noticed that the secreting lesions had higher parameters related to density; therefore, we can speculate that secreting lesions have higher densities in CT scans, as observed for cortisol-secreting adrenal adenomas [22]. In the differentiation between benign and malignant PCCs, the parameters that showed a significant difference were related to the homogeneity of the texture, with a more homogenous CT texture in malignant lesions.

We can speculate that one of the findings in our study (especially if combined with other description of radiomic use in adrenal masses [18,21]) is the possibility of a direct endocrine evaluation based on a CT report, obviously performed for a reason other than consideration of PCCs. The incidence of an incidental finding of a catecholamine-secreting adrenal mass (a PCC in an incidentaloma) is higher than the symptom-driven discovery of a secreting PCC [23]. Therefore, because the discovery of an adrenal incidentaloma is common during routine imaging, with frequency increasing with age [24], it is of utmost importance to have a tool such as that proposes in the present study in order to suggest an endocrine evaluation (i.e., plasma or urinary metanephrines) immediately at the first discovery of an adrenal mass. Although it is rare for a PCC to present with reduced lipid content, the most used threshold of 10 Hounsfield units is crossed not only by PCCs [2]

but also by lipid-poor adenomas [25] and in the case of autonomous cortisol secretion [22]. Therefore, lipid content should be used to indicate benign adrenal masses, and adrenal texture can be used to propose an endocrine assessment; a prospective study should be considered in such a scenario, as recently proposed in another independent cohort [20].

This function can be useful in clinical practice, as biochemical examination has shown high accuracy in detecting the presence of hormonal hypersecretion. Furthermore, texture analysis can be considered as an additional decision-making tool with respect to the first surgical approach for PCCs, i.e., whether to perform a laparoscopic or open surgery, owing to the higher probability of tumors with high PASS scores being malignant.

Artificial intelligence (AI) is an "umbrella" definition that usually indicates a computer program that can learn from a large amount of data and perform specific tasks [26]. The growth of deep learning machines, a successful technique for computer vision tasks, coupled with the availability of large imaging databases in recent decades, has led to the creation of multiple applications in medical imaging, especially to differentiate benign from malignant tissue. Researchers are exploring the far-reaching applications of AI in various clinical practices. The use of AI in CT analysis of adrenal nodules has been recently described [27] and could be very useful in two ways. First, in the case of identification of a nodule suspect of PCC (during abdominal imaging performed for another reason), texture analysis can be useful in identifying the presence of a hormonal hypersecretion, ensuring prompt referral of the patient to an endocrinologist (directly from the radiological report of the CT). Secondly, in the case of an already established diagnosis of secreting PCC (with evident catecholamine secretion), texture analysis can identify whether it is benign or malignant and facilitate the choice of surgical treatment.

Our study is subject to some limitations that should be notes. The main limitation is that it is a retrospective study, with a limited sample of patients analyzed. Our results should be considered a "proof of concept"; a larger, prospective, and multicentric study is necessary in order to confirm our initial finding in a larger cohort of patients. Late-phase, post-contrast CT acquisition (15 min) was available only in a few patients; therefore, a comparison was not reliable. Moreover, PCC is a rare condition; the number of patients in our cohort does not allow for further analysis, and we preferred a descriptive approach.

In conclusion, this can be considered a pilot study showing the applicability and utility of texture analysis in PCCs.

**Supplementary Materials:** The following supporting information can be downloaded at: https://doi.org/10.25430/researchdata.cab.unipd.it.00000741 (accessed on 25 December 2022).

**Author Contributions:** Data curation, F.T., M.I., and I.T.; Methodology, C.C.; Software, E.A. and A.T.; Writing—original draft, F.C. (Filippo Crimì), E.A., and A.T.; Writing—review and editing, C.S., E.Q., and F.C. (Filippo Ceccato). All authors have read and agreed to the published version of the manuscript.

**Funding:** This research received no external funding.

**Institutional Review Board Statement:** This study was conducted in accordance with the Declaration of Helsinki, and the Ethics Committee of Padova University Hospital (Comitato Etico per la Sperimentazione Scientifica) approved the study (protocol No. 85338-2022).

**Informed Consent Statement:** All subjects involved in the study signed an informed consent for the use of their data for scientific purposes.

**Data Availability Statement:** All data and results, with Supplementary Materials, are available in the repository.

**Conflicts of Interest:** The authors declare no conflict of interest.

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
