# Peer review of "CT Texture Analysis of Adrenal Pheochromocytomas: A Pilot Study"

_curroncol, doi:10.3390/curroncol30020167_

Round 1
Reviewer 1 Report
The paper name “CT Texture Analysis of Adrenal Pheochromocytomas: a Pilot Study” cover an interesting subject and is well written.
I will only have two recommendation to make for small addition to the paper.
First, a lot of texture analysis parameters are named but, it is lacking a short presentation of each parameters and to what it can be related ( it is mainly summarized in one sentence at lines 223 and 224).
Since AI is proving to be more useful everyday in many aspect of imaging few comments on this should be made.
Author Response
Reviewer 1
The paper name “CT Texture Analysis of Adrenal Pheochromocytomas: a Pilot Study” cover an interesting subject and is well written.
I will only have two recommendation to make for small addition to the paper.
First, a lot of texture analysis parameters are named but, it is lacking a short presentation of each parameters and to what it can be related (it is mainly summarized in one sentence at lines 223 and 224).
RE: Thank you for the suggestion, we have added a table in the supplementary material with the description of each textural parameter. We also added further description in the discussion section, where we tried to interpret the differences found among the different textural parameters.
Since AI is proving to be more useful everyday in many aspect of imaging few comments on this should be made.
RE: Thank you for raising the point. We have added a sentence in the discussion on that.

Reviewer 2 Report
Synopsis:
The authors present a semi-automated pipeline for classifying between secreting and not secreting pheochromocytomas (PCCs) in a single pilot study of custom Computed Tomography (CT) dataset (n=17). They report a correlation study of 1) Radiomics extracted features specifically (texture features) from an open access radiomics extraction software (LIFEx) with 2) malignancy score (PASS) of the adrenal gland and 3) secretion data with the aim of identification of malignant lesions and lesions that present hormonal hypersecretion.
Automated tumor staging is clinically relevant and is of interest to the target special issue of Machine Learning for Imaging-Based Cancer Diagnostics. However, the article has serious flaws in organization, readability, and analysis.
The language and writing quality of the manuscript are fair. However, the organization of the article is confusing and the innovation that it builds above the current state of the art is unclear.
The manuscript can be improved by considering the following comments:
General Comments:
1. Improve literature search: Missing details on use of Radiomics techniques. Texture features in CT radiomics have been used in the article and are part of standard workflow. Authors must report a survey of state-of-art in CT radiomics to address the problem of characterizing PCC’s. The authors state “only one paper by Ansquer et al [11] studied applications of PET/CT radiomics features in the characterization of PCCs and their genetic orientation”. However, following articles also address the problem in parts and should be cited, and the article's contribution must be situated in that light.
a. Yi, X., Guan, X., Zhang, Y. et al. Radiomics improves efficiency for differentiating subclinical pheochromocytoma from lipid-poor adenoma: a predictive, preventive and personalized medical approach in adrenal incidentalomas. EPMA Journal 9, 421–429 (2018).
b. Laderian, B., Ahmed, F. S., Zhao, B., Wilkerson, J., Dercle, L., Yang, H., ... & Fojo, A. T. (2019). Role of radiomics to differentiate benign from malignant pheochromocytomas and paragangliomas on contrast enhanced CT scans.
c. More recently
Mendi, R., & Gülbay, M. (2022). A CT Radiomics Analysis of the Adrenal Masses: Can We Discriminate Lipid-Poor Adenomas from the Pheochromocytoma and Malignant Masses?. Current Medical Imaging.
And many more
2. Improve the organization of the article: Currently there are a lot of places where the writing is incohesive. As an e.g., Page:5 contains Materials and Methods which should appear even before results are presented unless I am unaware of a different formatting that this journal follows. Similarly, the introduction should contain a brief summary of the contribution that the article achieves. Similarly, the conclusion is virtually non-existent. The abstract must contain a brief quantitative discussion of key results.
3. The authors must use edit the language of the manuscript/utilize language services. Currently a lot of casual phrases are used for eg:
a. Pg 5/7 “Anyway, this should be considered as a “proof-of concept” study,”.
b. Rephrase on Pg 4 “the most stringent gain of our study”
c. I am not sure what this is supposed to mean “Regarding the histopathological characteristics, the study showed a significative correlation, yet weak”.
4. Table S1 is referred on Page 2 but is missing in the uploaded version. Does it refer to the one in link (10.25430/researchdata.cab.unipd.it.00000741). Clarify?
Specific Comments:
1. My biggest grievance of the manuscript is a lack of description as to why this study was conducted only with radiomics texture features? LIFEx the open-source image analysis software used in this study and many others (Pyradiomics) allows several other first/second order statistics features to be extracted! Why did the authors restrict themselves to only texture features.
2. Did the authors consider a sensitivity analysis of the features (feature importance study) that will chart dominant features based on some statistical metric (such as mean decrease in impurity) that drive classification outcome?
3. Contributions of the article are unclear as discussed in Synopsis; the above points must be clearly addressed.
4. A standard practice for such a study is to visualize an ensemble of measures and then decide on important features which is missing from the discussion.
5. It is essential that an imbalanced dataset such as this be split taking into consideration the minority class. The current discussion does not mention how the authors tackle the minority secretive class. Furthermore, the train-validation-test distributions must be explicitly mentioned in a separate section under methods.
6. Do the author's mean in Figure 1 and Figure 2 mean “unenhanced scan (A), arterial and venous scan (B and C).”? Currently, for Figure 1 they mention “unenhanced scan (A) and in venous scans (B-H).” Clarify?
Summarily, the authors report the use of Radiomics texture features for distinguishing secreting and not secreting PCCs in CT scans. Comparisons with SOTA are limited, and the manuscript should be revisited to incorporate the motivation/improvement against thereof.
However, CT radiomics on PCCs are not widely studied, especially one that aims to distinguish hyper/non-secretors and is therefore of interest to the community. However, a more robust presentation of the results is necessary, and therefore the manuscript in its current form will benefit from a major revision.
Author Response
Reviewer 2
Synopsis:
The authors present a semi-automated pipeline for classifying between secreting and not secreting pheochromocytomas (PCCs) in a single pilot study of custom Computed Tomography (CT) dataset (n=17). They report a correlation study of 1) Radiomics extracted features specifically (texture features) from an open access radiomics extraction software (LIFEx) with 2) malignancy score (PASS) of the adrenal gland and 3) secretion data with the aim of identification of malignant lesions and lesions that present hormonal hypersecretion.
Automated tumor staging is clinically relevant and is of interest to the target special issue of Machine Learning for Imaging-Based Cancer Diagnostics. However, the article has serious flaws in organization, readability, and analysis.
The language and writing quality of the manuscript are fair. However, the organization of the article is confusing and the innovation that it builds above the current state of the art is unclear.
RE: Thank you for the suggestion: according to yours and other reviewers’ comments we highlighted the novelty of the paper, and the English was reviewed by a naïve.
The manuscript can be improved by considering the following comments:
General Comments:
- Improve literature search: Missing details on use of Radiomics techniques. Texture features in CT radiomics have been used in the article and are part of standard workflow. Authors must report a survey of state-of-art in CT radiomics to address the problem of characterizing PCC’s. The authors state “only one paper by Ansquer et al [11] studied applications of PET/CT radiomics features in the characterization of PCCs and their genetic orientation”. However, following articles also address the problem in parts and should be cited, and the article's contribution must be situated in that light.
- Yi, X., Guan, X., Zhang, Y. et al. Radiomics improves efficiency for differentiating subclinical pheochromocytoma from lipid-poor adenoma: a predictive, preventive and personalized medical approach in adrenal incidentalomas. EPMA Journal 9, 421–429 (2018).
- Laderian, B., Ahmed, F. S., Zhao, B., Wilkerson, J., Dercle, L., Yang, H., ... & Fojo, A. T. (2019). Role of radiomics to differentiate benign from malignant pheochromocytomas and paragangliomas on contrast enhanced CT scans.
- More recently
Mendi, R., & Gülbay, M. (2022). A CT Radiomics Analysis of the Adrenal Masses: Can We Discriminate Lipid-Poor Adenomas from the Pheochromocytoma and Malignant Masses?. Current Medical Imaging.
And many more
RE: thank you for the suggestion. We have improved as suggested the introduction and the discussion taking into account these papers.
- Improve the organization of the article: Currently there are a lot of places where the writing is incohesive. As an e.g., Page:5 contains Materials and Methods which should appear even before results are presented unless I am unaware of a different formatting that this journal follows. Similarly, the introduction should contain a brief summary of the contribution that the article achieves. Similarly, the conclusion is virtually non-existent. The abstract must contain a brief quantitative discussion of key results.
RE: we modified the manuscript accordingly.
- The authors must use edit the language of the manuscript/utilize language services. Currently a lot of casual phrases are used for eg:
- Pg 5/7 “Anyway, this should be considered as a “proof-of concept” study,”.
- Rephrase on Pg 4 “the most stringent gain of our study”
- I am not sure what this is supposed to mean “Regarding the histopathological characteristics, the study showed a significative correlation, yet weak”.
RE: we modified the manuscript accordingly.
- Table S1 is referred on Page 2 but is missing in the uploaded version. Does it refer to the one in link (10.25430/researchdata.cab.unipd.it.00000741). Clarify?
RE: we added a direct link for supplementary material.
Specific Comments:
- My biggest grievance of the manuscript is a lack of description as to why this study was conducted only with radiomics texture features? LIFEx the open-source image analysis software used in this study and many others (Pyradiomics) allows several other first/second order statistics features to be extracted! Why did the authors restrict themselves to only texture features.
RE: Thank you for rising the point. LIFEx extracts both first and second order statistics, hence, for clarity we have specified it into the materials and methods section.
- Did the authors consider a sensitivity analysis of the features (feature importance study) that will chart dominant features based on some statistical metric (such as mean decrease in impurity) that drive classification outcome?
RE: Thank you for the question. We did not perform such analysis since we did not train a model for PCCs classification. Indeed, having only 17 patients with all these data we preferred to perform only a descriptive analysis of textural parameters in the cohort. The same reply is useful also for question 5: we added and highlighted in the limitation of the study that the limited cohort does not allow further analysis (pheochromocytoma is a rare disease)
- Contributions of the article are unclear as discussed in Synopsis; the above points must be clearly addressed.
RE: we modified the manuscript accordingly.
- A standard practice for such a study is to visualize an ensemble of measures and then decide on important features which is missing from the discussion.
RE: Thank you for the question. As stated before, we did not perform a feature importance study. We added in the discussion section a sentence where we tried to interpret the differences found among the different textural parameters
- It is essential that an imbalanced dataset such as this be split taking into consideration the minority class. The current discussion does not mention how the authors tackle the minority secretive class. Furthermore, the train-validation-test distributions must be explicitly mentioned in a separate section under methods.
RE: Thank you, in order to overcome multiple testing biases, we used the Bonferroni correction in the statistical analysis (we selected only those feature that showed a significant p value in both Mann-Whitney and Bonferroni texts). We did not perform a train-validation-test due to the small sample of the cohort.
The authors should correct for multiple testing as the same features were used for calculation of
sensitivity, specificity and accuracy for individual and combined analysis
RE: Thanks for raising the point. We did not write it in the text but we used a correction for
the Wilcoxon test results with Bonferroni and we selected only those feature that showed a
significant p value in both.
- Do the author's mean in Figure 1 and Figure 2 mean “unenhanced scan (A), arterial and venous scan (B and C).”? Currently, for Figure 1 they mention “unenhanced scan (A) and in venous scans (B-H).” Clarify?
RE: Thank you for rising the point. We apologize since the text was not clear enough. We did not obtain parameters that were significantly different among those extracted from CT images in arterial phase. For clarity, we rephased the results section.
Summarily, the authors report the use of Radiomics texture features for distinguishing secreting and not secreting PCCs in CT scans. Comparisons with SOTA are limited, and the manuscript should be revisited to incorporate the motivation/improvement against thereof.
RE: Thank you for the observation. Indeed, nuclear medicine imaging is surely more accurate than CT in identifying secreting and not secreting PCCs, however we think that in a setting where the radiologist identifies a lesion suspect for PCCs at a CT examination the texture could be a valid tool to further address the patient to an Endocrinological Consult. Meanwhile, for the identification of benignity of malignity of a PCCs at the best of our knowledge for the SOTA imaging the only real parameter that helps distinguishing between these forms is the presence of nodal locoregional or distant metastases.
However, CT radiomics on PCCs are not widely studied, especially one that aims to distinguish hyper/non-secretors and is therefore of interest to the community. However, a more robust presentation of the results is necessary, and therefore the manuscript in its current form will benefit from a major revision.

Reviewer 3 Report
This manuscript deals with an interesting and timely topic. Main weaknesses are limited sample of patients and consequently lack of validation.
More specific comments:
1) Material and methods: Could you please add a table with clinical characteristics of patients?
Did you try to integrate HU values of the lesion obtained from unenhanced CT with radiomics data?
Did you perform a late phase post-contrast CT acquisition (15 min)?
2) Results: Why didn’t you report relevant textural features obtained from arterial phase?
3) Discussion: please comment in a more detailed way relevant results obtained from the other similar studies with respect to your own findings
Author Response
Reviewer 3:
This manuscript deals with an interesting and timely topic. Main weaknesses are limited sample of patients and consequently lack of validation.
More specific comments:
1) Material and methods: Could you please add a table with clinical characteristics of patients?
Did you try to integrate HU values of the lesion obtained from unenhanced CT with radiomics data?
Did you perform a late phase post-contrast CT acquisition (15 min)?
RE: thank so much for the suggestions. We have added a table with a description of the patients reported (table 1). All data from CT (among them also HU) have been measured and reported in the supplementary material (it is a very large table, with a large number of results…radiomic is an analysis of big data). In the result and discussion, we reported the most significant findings, nonetheless all results are available. A late phase was available only in few patients, therefore it has not been considered: it is a limitation of the paper and we added it in the dedicated section.
2) Results: Why didn’t you report relevant textural features obtained from arterial phase?
RE: because they were not significant. All results, also those obtained from arterial phase (page 10-14 of the table), are available in supplementary repository data. We decided not to describe them in the paper.
3) Discussion: please comment in a more detailed way relevant results obtained from the other similar studies with respect to your own findings
RE: we added more data in similar studies, thanks for the suggestion.

Round 2
Reviewer 1 Report
Nice work, thank you for the edit.
Reviewer 2 Report
My concerns are mostly addressed. A lot of language in the paper is moved to discussion, which I believe is most suited to be in the Introduction. All prior work must be referenced in Introduction, while the discussion is the place for the author's presented work.
Author Response
REVIEWER 2:
My concerns are mostly addressed. A lot of language in the paper is moved to discussion, which I believe is most suited to be in the Introduction. All prior work must be referenced in Introduction, while the discussion is the place for the author's presented work.
RE: Thank so much for the careful revision, that improved in a substantial way our paper. We changed the discussion, moving the prior works that were useful for readers in the introduction and with a mention in the discussion, with the comparison with our work. Some paper remained in the discussion, in order to create an homogeneous flow.
Reviewer 3 Report
It's ok for me.